# Molecular Responses to High Hydrostatic Pressure in Eukaryotes: Genetic Insights from Studies on *Saccharomyces cerevisiae*

**DOI:** 10.3390/biology10121305

**Published:** 2021-12-09

**Authors:** Fumiyoshi Abe

**Affiliations:** Department of Chemistry and Biological Science, College of Science and Engineering, Aoyama Gakuin University, Sagamihara 252-5258, Japan; abef@chem.aoyama.ac.jp; Tel.: +81-42-759-6233; Fax: +81-42-759-6511

**Keywords:** yeast, *Saccharomyces cerevisiae*, high-pressure response, genetic manipulation, transcriptomics, piezophysiology

## Abstract

**Simple Summary:**

High hydrostatic pressure generally has an adverse effect on the biological systems of organisms inhabiting lands or shallow sea regions. Deep-sea piezophiles that prefer high hydrostatic pressure for growth have garnered considerable scientific attention. However, the underlying molecular mechanisms of their adaptation to high pressure remains unclear owing to the challenges of culturing and manipulating the genome of piezophiles. Humans also experience high hydrostatic pressure during exercise. A long-term stay in space can cause muscle weakness in astronauts. Thus, the human body indubitably senses mechanical stresses such as hydrostatic pressure and gravity. Nonetheless, the mechanisms underlying biological responses to high pressures are not clearly understood. This review summarizes the occurrence and significance of high-pressure effects in eukaryotic cells and how the cell responds to increasing pressure by particularly focusing on the physiology of *S. cerevisiae* at the molecular level.

**Abstract:**

High hydrostatic pressure is common mechanical stress in nature and is also experienced by the human body. Organisms in the Challenger Deep of the Mariana Trench are habitually exposed to pressures up to 110 MPa. Human joints are intermittently exposed to hydrostatic pressures of 3–10 MPa. Pressures less than 50 MPa do not deform or kill the cells. However, high pressure can have various effects on the cell’s biological processes. Although *Saccharomyces cerevisiae* is not a deep-sea piezophile, it can be used to elucidate the molecular mechanism underlying the cell’s responses to high pressures by applying basic knowledge of the effects of pressure on industrial processes involving microorganisms. We have explored the genes associated with the growth of *S. cerevisiae* under high pressure by employing functional genomic strategies and transcriptomics analysis and indicated a strong association between high-pressure signaling and the cell’s response to nutrient availability. This review summarizes the occurrence and significance of high-pressure effects on complex metabolic and genetic networks in eukaryotic cells and how the cell responds to increasing pressure by particularly focusing on the physiology of *S. cerevisiae* at the molecular level. Mechanosensation in humans has also been discussed.

## 1. General Effects of High Hydrostatic Pressure on Biological Systems

While high hydrostatic pressure is a commonly known characteristic of deep-sea environments, the human body also experiences high pressure. However, it is important to distinguish between the isostatic pressure acting equally in all directions and uniaxial stress. While deep-sea organisms are constantly exposed to the isostatic pressure, uniaxial (or directional) pressure can act on human bones. This review primarily focuses on biological responses to isostatic hydrostatic pressure. A hydrostatic pressure of 10 MPa or higher generally has an adverse effect on the biological systems of organisms inhabiting lands or shallow sea regions [1,2]. Hydrostatic pressure increases by 0.1 MPa (0.1 MPa = 1 bar = 0.9869 atm; for clarity, “MPa” is used throughout) for every 10-m depth. Therefore, deep-sea organisms are exposed to a high pressure of 110 MPa in Challenger Deep, which is the deepest point in the Marina Trench (10,900 m). Deep-sea microorganisms or piezophiles that prefer high hydrostatic pressure for growth have garnered considerable scientific attention [2,3,4,5]. However, the molecular mechanism underlying the high-pressure adaptation has not been elucidated in piezophiles because of the difficulties in their cultivation and genetic manipulation (e.g., gene disruption, overexpression, or mutagenesis). Humans also experience high hydrostatic pressure when hip joints are exposed to a pressure of 18 MPa during exercise [6]; the back of the teeth is also exposed to the same levels of pressure. A long-term stay in space can cause muscle weakness in astronauts. Thus, the human body indubitably senses mechanical stresses such as hydrostatic pressure and gravity. Nonetheless, the mechanisms underlying biological responses to high pressures are not clearly understood. Figure 1 illustrates pressure ranges, research fields, and the main subjects in high-pressure bioscience and biotechnology.

Approximate pressure ranges affecting various biological processes are listed in Table 1 [7]. Generally, microorganisms can survive at high hydrostatic pressures in the range of several dozen MPa; a pressure higher than 200 MPa is lethal to most microorganisms but not to spores of *Clostridium* species, which are resistant to pressures greater than 400 MPa. The non-thermal sterilization of food materials using ultra-high pressure has been studied previously [8,9]. The effects of high pressure depend on the magnitude, pressurizing periods, temperature, pH, oxygen availability, and nutrient composition, which are complex and difficult to interpret. Oxygen supply is one of the limiting factors to cultivating aerobic organisms in a closed hydrostatic chamber. Yeast species isolated from mud samples obtained through submersible explorations with SHINKAI 6500 at the Japan Trench (~6500 m depth) were strictly aerobic, probably because of the oxygen-rich environment and scarcity of fermentable sugars such as glucose in the deep-sea. *Saccharomyces cerevisiae* is a facultative anaerobe, which makes it a useful model organism for studying the effects of high hydrostatic pressure. The *S. cerevisiae* genome encodes 6611 genes (*Saccharomyces* Genome Database, Genome Snapshot: https://www.yeastgenome.org/genomesnapshot; accessed on 8 December 2021). Global functional screening using a *S. cerevisiae* gene knockout mutant library revealed many unexpected genes and intracellular pathways that are associated with environmental stress resistance [10,11,12]. Figure 2 illustrates the bioinformatic tools employed to elucidate the mechanisms of high-pressure responses and adaptation in *S. cerevisiae*, as a model organism.

In any reaction, the application of pressure yields a fundamental physical parameter, the volume change. The following two equations describe the effect of hydrostatic pressure on equilibrium A ⇌ B and reaction A → B, respectively:(∂ln*K*/∂*p*)_T_ = −Δ*V*/*RT*
(1)
(∂ln*k*/∂*p*)_T_ = −Δ*V*^≠^/*RT*
(2)
where *K* is the equilibrium constant, *k* is the rate constant, *p* is the pressure (MPa), *T* is the absolute temperature (K), *R* is the gas constant (mL·MPa·K^−1^·mol^−1^), Δ*V* is the difference between final and initial volumes in the entire system at equilibrium (reaction volume) including the solute and the solvent, and Δ*V*^≠^ is the apparent volume change in activation (activation volume), representing the difference between the volumes of the ground and transition states. The effects of high pressure depend on the sign and amplitude of the change in volume associated with any reaction; thus, an increase in volume promotes inhibition of the reaction due to a pressure increase, and vice versa. While temperature accelerates any reaction, per the Arrhenius equation, pressure accelerates, inhibits, or does not affect reactions depending on the Δ*V* and Δ*V*^≠^ values. Although the thermodynamic law is straightforward, estimating whether the expression of individual genes and levels of proteins increase or decrease when living cells are exposed to high pressure is difficult. Nevertheless, many informative studies have reported the biophysical effects of high pressure on biological machinery. In general, a pressure approximately greater than 100 MPa dissociates multimeric proteins as the hydration of charged groups and exposure of nonpolar groups to water is usually accompanied by negative volume changes [13,14]. Therefore, dissociated forms of proteins are formed in the aqueous solution under high pressure. Pressures greater than 60 MPa cause dissociation of ribosome subunits in *Escherichia coli* (at least in part), thereby limiting their growth [13]. Furthermore, the level of eukaryotic elongation factor-2 is decreased under continuous high-pressure culture at 30 MPa in HeLa carcinoma and T/C28a4 chondrocyte cell lines, suggesting that, at least in part, downregulated elongation factor-2 is attributed to the attenuation of protein synthesis under high pressure [15].

A deep-sea bacterium *Moritella profunda* optimally grows under 20–24 MPa at 6 °C under a laboratory condition [16] and is, thus, a psychrophillic piezophile. Interestingly, the dihydrofolate reductase of this bacterium is optimally active at 50 MPa and its secondary structure is stable up to 80 °C [17]. Therefore, a microbial growth profile does not always reflect intracellular enzymatic properties. Indeed, numerous studies have revealed that many deep-sea animal proteins intrinsically exhibit less sensitivity to high pressure than their orthologs from shallow species. Presumably, these differences reduce volume changes occurring in reactions [18]. For example, the formation of cytoskeletal actin filaments is more resistant to high pressure in deep-sea fish species than in their shallow sea counterparts [19]. The acquisition of this pressure tolerance in actin occurs, in part, due to the presence of more salt-bridges between amino acids associated with ATP-binding and structural stability [19]. It has been well-documented that trimethylamine *N*-oxide (TMAO) is the key osmolyte in marine fish, which can effectively counteract the inhibitory effects induced by high pressure on numerous proteins. Accordingly, TMAO content in marine fish increases with depth of capture (see Reviews: [20,21,22]). In fact, recent whole genome sequencing of a snailfish from the Yap Trench (~7000 m depth) revealed the presence of five copies of a gene encoding a flavin-containing monooxygenase-3 that catalyzes trimethylamine (TMA) to TMAO. In the fish, TMA can be supplied from gut bacteria [23].

Although high pressure induces the unfolding of protein monomers [14,24,25], the effect elicited by pressure is highly dependent on the actual objects and temperature within the range of a few hundred MPa to 1 GPa. Meanwhile, GFP and heat shock proteins are stable even above 1 GPa [26,27]. During protein unfolding, water molecules penetrate the cavities within the proteins and nonpolar groups are exposed to the solvent [28]. Consequently, proteins form aggregates within the cell that are harmful to cells.

The phase transition of lipid bilayers in biological systems is highly sensitive to high pressure. High pressure and low temperature reorder acyl chains of phospholipids, making the membrane stiffer [29,30]. In dipalmitoylphosphatidylcholine lipid bilayers, the temperature for the transition (*T*_m_) from the ripple gel (P_β_’) phase to the liquid crystalline (L_α_) phase increases by 24 °C with an increase in pressure of 100 MPa [31]. Although a clear phase transition is not observed in natural membranes in living cells, high pressures indubitably harden the membranes, which in turn affects the functionality and structure of transmembrane proteins. The occurrence and response of *S. cerevisiae* cells to high pressure are described in the following sections.

## 2. Effects of High Pressure on Cultured Human Cells and Tissues

The meniscus is a “C” shaped cartilage-like tissue located inside and outside of the knee joint that absorbs shock and stabilizes the knee joint. However, intense physical exercise and aging can damage the meniscus. The force applied to the meniscus is anisotropic and is complicated by the addition of shear stress, thus making molecular quantitative analysis of the cellular responses difficult. Hydrostatic pressure can regulate the metabolic activities of meniscal cells. For instance, the static application of 4 MPa pressure for 4 h suppresses the transcription of MMP-1 and MMP-13 genes in human meniscal cells in alginate beads [32]. In contrast, cyclic hydrostatic pressure at 1 Hz upregulates genes encoding type I collagen, TIMP-1, and TIMP-2. Meanwhile, upon explant, rabbit meniscal cells increase the transcription of genes encoding MMP-1, MMP-3, TIMPs, NOS2, COX2, IL-1β, and IL-6 [33]. However, the cyclic application of pressure of 1 MPa at 0.5 Hz for 4 h blocked the culture-induced increases in catabolic gene expression. Collectively, these studies indicate that mechanical loading on the meniscus can modulate the physiology of meniscal cells at the transcriptional level to maintain homeostasis in the meniscal tissue during culture [33].

Osteoarthritis is a disease in which the cartilage between the joint bones deteriorates and causes pain, swelling, and ultimately joint deformation. As per statistics, 10 million people of more than 50 years of age in Japan experience knee pain due to knee osteoarthritis. Attempts have been made to solve these problems using regenerative medicine based on cells cultured under high pressure. A pressure of 1 to 15 MPa was used to differentiate between the mesenchymal stem cells isolated from the human body and chondrocytes, which were reviewed by Elder et al. [34] and Pattappa et al. [35]. They tried to determine how the applied pressure leads to the upregulation of specific genes inside the cell. Identifying the sensor molecule that first senses hydrostatic pressure in the cell was difficult. Various candidates for sensor proteins in the induction of chondrogenesis, such as estrogen receptors, voltage-gated ion channels, G-protein-coupled receptors, integrin α10β1, and Ca^2+^ signaling pathways via TRPV4 channels have been reported. Nevertheless, the molecular basis of pressure-induced activation has not been established [35]. As unique applications in medical engineering, hydrostatic pressures at levels much higher than those found in physiological conditions were used to disinfect bone, tendons, and cartilage. The administration of short-term high pressure at 600 MPa to resected bone segments immediately after surgery offers an alternative to the conventional cancerous bone treatment. Under this condition, normal and malignant cells are irreversibly damaged, thereby efficiently blocking the outgrowth of cells from cancerous bone and cartilage segments (see Review [36]). Moreover, high hydrostatic pressure treatment at 480 MPa for 10 min has been applied to devitalize human cartilage with the goal of inducing subsequent cultivation of chondrocytes and mesenchymal stem cells on the devitalized tissue [37]. Meanwhile, the treatment of skin tissues at a moderate pressure of 50 MPa for more than 36 h induces cell death via apoptosis. Subsequent in vivo grafting of the apoptosis-induced inactivated skin was successful. This method is thought to have an advantage over inducing complete cell death via necrosis, as apoptotic cells do not generally promote inflammation [38]. Future progress can be expected for feasible applications of high pressure in medical engineering.

## 3. Effects of Lethal Levels of High Pressure on Yeast Survival

Figure 3 illustrates the occurrences and responses of *S. cerevisiae* cells upon exposure to various levels of hydrostatic pressure. In a pioneering study, Rosin and Zimmerman reported that when a pressure of 96.6 MPa is applied for 4 h, the occurrence of cytoplasmic petite mutants increases, which are characterized by a small colony size and respiration deficiency and reflect the high-pressure sensitivity of mitochondrial functions [39]. Pressures in the range of 100–150 MPa cause disruption of the spindle pole bodies and microtubules [40], and those of 200–250 MPa induce tetraploids and homozygous diploids [41]. A moderate heat treatment (e.g., 42 °C for 30 min) dramatically increases the survival rate after a subsequent heat-mediated toxic treatment (e.g., 50 °C for 10 min). Among heat shock proteins (Hsps), the molecular chaperone Hsp104 plays a primal role in this acquired heat tolerance mechanism by facilitating the unfolding of the denatured intracellular proteins [42]. This moderate heat treatment enhances cell survival against lethal levels of high pressure at 140–180 MPa with a 100–1000-fold increase in viability [43]. After 140 MPa pressure treatment, Hsp104 associate with insoluble protein aggregates, suggesting the contribution of Hsp104 contributes for the unfolding of high pressure-induced denatured proteins [44,45]. The *hsp104*Δ mutant did not acquire heat-inducible high-pressure tolerance (hereafter referred to as piezotolerance; [44]. Iwahashi et al. reported that piezo-tolerant mutants surviving at 180 MPa for 1 h (25 °C) also display resistance to 1% H_2_O_2_ for 60 min (0 °C) [46]. Moreover, Palhano et al. reported that treatment of cells with a moderate concentration of H_2_O_2_ (0.4 mM [0.0012%], 45 min) or 6% ethanol can increase the piezotolerance of yeast cells at 220 MPa for 30 min [47]. H_2_O_2_ is known to induce many antioxidant genes, including *GSH1* (which encodes γ-glutamylcysteine synthetase, an enzyme in glutathione (GSH) biosynthesis). The addition of GSH (>1 mM) increases piezotolerance in *S. cerevisiae* [47]. Meanwhile, ethanol stress upregulates genes involved in energy metabolism, protein destination, and stress tolerance including the cytoplasmic catalase T (Ctt1) and mitochondrial superoxide dismutase Sod2. *CTT1* and *SOD2* are also upregulated by high pressure [48]. A deletion mutant for *COX1* encoding the subunit I of cytochrome *c* oxidase in mitochondria is also sensitive to pressure at 200 MPa [49]. Taken together, these findings indicate that high hydrostatic pressure of approximately 200 MPa is likely to exert oxidative stresses in yeast cells, and the cellular defense systems against high pressure, at least in part, converge on oxidative stress responses for “cell survival.” Consistently, our recent study indicated that the scavenging activity of superoxide anion O_2_^•–^ by superoxide dismutase 1 (Sod1) is required for “cell growth” under a moderate growth-permissive pressure at 25 MPa [50]. Moreover, high pressure promoted the accumulation of O_2_^•–^ in the mitochondrial inner space and the cytoplasm. Meanwhile, mutations in Sod1 that compromise the scavenging activity (or *CCS1* deletion), encoding a molecular chaperone that delivers copper to Sod1, cause deficient growth under 25 MPa [50].

Pressure pre-treatment at a sublethal level (50 MPa for 1 h) increases the viability of cells at 200 MPa [51]. This acquired piezotolerance via moderate pressure treatment is governed by two transcription factors, namely, Msn2 and Msn4, which are induced by various stresses. The loss of both *MSN2* and *MSN4* genes results in susceptibility to high pressure [51]. *HSP12* is a well-known target of Msn2/Msn4. In fact, the application of a pressure of 50 MPa increases the *HSP12* transcript level in wild-type *S. cerevisiae* cells but does not in the *msn2*∆*msn4*∆ mutant [51]. However, the single deletion of *HSP12* did not decrease cell viability at 125 MPa for 1 h [52]. Therefore, a certain set of genes, under the control of Msn2/Msn4, could be required for the development of piezotolerance in yeast. Other genes highly upregulated by pressure at 25 MPa, such as the *DAN/TIR* family [53], that encode cell wall mannoproteins are not under the control of Msn2/Msn4 (see below). How high pressure regulates Msn2 and Msn4 after the transcription of their downstream genes remains unknown. Trehalose is a nonreducing disaccharide that protects proteins, membranes, and other macromolecules against various stresses. Trehalose also plays a role in the piezotolerance of yeast by preventing the formation of protein aggregates and promoting the refolding of Hsp104. Trehalose is hydrolyzed by the neutral trehalase Nth1, generating two glucose molecules from one trehalose molecule. Yeast mutants lacking Nth1 are susceptible to high pressures [45]. Neutral trehalase is required for recovering from the damage after high-pressure treatments, whereas it is dispensable for cell survival under high pressure. Accordingly, the recovery process from pressure-induced damage in yeast requires glucose at atmospheric pressure.

The composition of fatty acids affects the survival of the cell under lethal pressure of 150–200 MPa. A deletion mutant of *OLE1* encoding a membrane-bound Δ9 desaturase was cultured in a medium supplemented with various fatty acids such as palmitoleic acid (C16:1), oleic acid (C18:1), linoleic acid (C18:2), and linolenic acid (C18:3). Subsequently, the cells were exposed to pressures of 150–200 MPa for 30 min. The effect of increasing cell viability was in the order of linolenic acid > linoleic acid > oleic acid > palmitoleic acid. Therefore, a higher proportion of unsaturated fatty acids is likely to contribute to maintaining appropriate membrane fluidity under high pressure and reduced membrane fluidity would be fatal under high pressure [54]. However, reduced membrane fluidity is not the sole reason for cell death at 150–200 MPa because low temperatures (0–4 °C) are not lethal for the cells.

## 4. Tryptophan Uptake Is Crucial for Yeast Physiology under High Pressure

The hydrostatic pressure of 15 to 25 MPa is relatively mild for microorganisms and most of them can grow despite their decreased growth rate. The growth properties of experimental yeast strains differ greatly under high pressure and are dependent on the strain’s requirement for tryptophan from the external medium. Experimental *S. cerevisiae* strains usually exhibit nutrient auxotrophic markers (e.g., *ade2, ura3, his3, lys3, leu2*, and *trp1*) for plasmid selection, that is, these strains require some elements (e.g., adenine, uracil, histidine, lysine, leucine, and tryptophan) for growth from their external medium. The auxotrophic characteristics of experimental yeast strains are similar to animals, including humans, because they also require histidine, lysine, leucine, and tryptophan from their diet. Tryptophan auxotrophs (Trp^−^) are highly sensitive to high pressures. Trp^−^ cannot grow under 15–25 MPa, whereas tryptophan prototrophs (Trp^+^) that can synthesize tryptophan, can grow under the same level of pressure [55,56]. Trp^−^ strains also exhibit growth deficiency at low temperatures of 10–15 °C, which is consistent with an earlier report [57]. The high-pressure sensitivity is attributed to properties specific to tryptophan uptake, which in *S. cerevisiae* is mediated by the tryptophan permeases Tat1 and Tat2: (i) the transport activity of Tat1 and Tat2 is readily impaired if the membrane is ordered either by high pressure or low temperature; and (ii) Tat1 and Tat2 undergo degradation via ubiquitination under high pressure (see below). Excessive addition of tryptophan or overexpression of *TAT1* or *TAT2* enables Trp^−^ cells to grow under 25 MPa [55,56]. Accordingly, *S. cerevisiae* can grow at pressures of up to 25 MPa if tryptophan is available in the external medium.

## 5. High Pressure Induces Degradation of Tryptophan Permeases via Ubiquitination

Ubiquitin is a small protein consisting of 76 amino acid residues and is highly conserved among eukaryotes. Ubiquitin covalently binds to the lysine residue of the target protein to be degraded via the ubiquitin-activating enzyme (E1), ubiquitin-conjugating enzyme (E2), and ubiquitin ligase (E3). The ubiquitinated protein is then transported to the 26S proteasome or vacuole (lysosome in animal cells) for degradation [58,59]. Deficiency in the ubiquitin system causes human diseases such as developmental anomalies, cancer, or neurological disorders; thus, clearance of unwanted proteins by ubiquitination is crucial. Hydrostatic pressure leads to the structural perturbation of biological membranes, which directly or indirectly affects the membrane proteins. As mentioned above, Trp^−^ strains are sensitive to moderate pressures of 15–25 MPa. Our research group has isolated many mutant strains that acquired the ability to grow under high pressure from the tryptophan-requiring strain YPH499 (a strain that cannot grow at 25 MPa) [56]. Genetic analysis showed that one of these mutations, *HPG1*, occurs in the catalytic domain of the E3 enzyme Rsp5 (Figure 3). E3 plays an important role in the target recognition, and Tat2 is a substrate for Rsp5 ubiquitin ligase. Therefore, Tat2 levels were high in the plasma membrane of the *HPG1* mutant, allowing sufficient tryptophan uptake by the cell, whereas Tat2 was degraded in the wild-type strain. *HPG2* mutation sites are located within the cytoplasmic tails of Tat2 [60], which are accompanied by the loss of negative charge within the cytoplasmic tail. Therefore, the negatively charged amino acid residues in the cytoplasmic tails may be required for Tat2 to interact with the Rsp5 complex via ionic interactions. However, how the denatured states of Tat1 and Tat2 can be structurally characterized and how denatured proteins are recognized by the Rsp5 complex remain to be elucidated.

## 6. Transcriptional Analysis of Genes Responsive to High Pressures

The yeast genome consists of approximately 12 million base pairs that encode 6611 genes. Among them, the functions of 5229 genes are known or predicted, whereas the rest are poorly characterized (*Saccharomyces* Genome Database). In addition to house-keeping genes that are essential for survival (such as energy metabolism and cell division), many genes are transcriptionally induced when the cells are exposed to critical situations such as heat or oxidative stress. Transcriptionally induced genes are necessary for establishing a cellular defense system in adverse conditions. DNA microarray hybridization and the more recent RNA-Seq techniques are widely used to comprehensively investigate the transcriptional level in response to environmental changes. Under the pressure of 30 MPa and a temperature of 25 °C, which allows for the growth of tryptophan prototrophs, 366 genes were upregulated by more than 2-fold, and 253 genes were downregulated by more than 2-fold [61]. According to the functional categories of the MIPS database, homology data, and yeast genome, the highly upregulated genes were essential for cell cycle, DNA processing, cell rescue, defense and virulence, and metabolism. Heat shock-responsive genes including *HSP12* and *HSP104* are induced by high pressure during acquired piezotolerance [61]. In our DNA microarray analysis, gene expression was compared under conditions of atmospheric pressure (0.1 MPa, 24 °C), high pressure (25 MPa, 24 °C), and low temperature (0.1, 15 °C), and found that the *DAN/TIR* family genes that encode mannoprotein of the cell wall were highly upregulated. Cells pre-exposed to high pressure or low temperature acquired tolerance when treated with moderate concentrations of SDS and zymolyase or a lethal level of high pressure because mannoproteins maintain cell wall integrity (125 MPa for 1 h) [53]. The *DAN/TIR* family genes are significantly induced under hypoxic conditions [62]. Therefore, intracellular signaling pathways responsive to high pressure, low temperature, and hypoxia might interact to establish defense systems.

## 7. Global Functional Analysis of Genes Required for Growth under High Pressure

Among the 6611 genes encoded by the *S. cerevisiae* genome, the single deletion of approximately 4800 genes is not lethal under normal culture conditions. These genes are considered to contribute to efficient cell growth or survival under hostile environmental conditions; therefore, some of them are functionally redundant. PCR-based systematic deletion of genes was performed (http://www-sequence.stanford.edu/group/yeast_deletion_project/project_desc.html#intro; accessed on 8 December 2021) [63], and deletion (gene knockout) libraries are available for purchase.

Using the deletion library, mutations causing hypersensitivity to high pressure (25 MPa and 24 °C) or low temperature (0.1 MPa and 15 °C) were identified within 4828 non-essential genes. To the best of my knowledge, this is the only study to perform a large-scale screening of the yeast deletion library to identify genes required for growth under a moderate high pressure. This analysis revealed 84 genes, of which 75 were found to be required for growth at 25 MPa and 24 °C and 57 were found to be required for growth at 0.1 MPa and 15 °C [64,65]. There was a marked overlap of 48 genes, implying that various biological functions have common roles allowing cell growth under high pressure and low temperature [64]. These 84 genes were classified into biological processes using STRING, a tool for functional enrichment analyses (https://string-db.org; accessed on 8 December 2021) (Table 2) [66]. Interestingly, there are several distinct clusters in physical or functional protein-protein interactions among the 84 proteins predicted on STRING (Figure 4). Therefore, these specific cellular functions must be highly important for growth under high pressure and low temperature. The 84 deletion mutants displayed variable sensitivity toward high pressure and low temperature. Consistent with earlier observations of growth deficiency in Trp^−^ strains [55,56], aromatic amino acids are crucial for high-pressure and low-temperature growth. Indeed, it has also been reported that mutants defective in tryptophan biosynthesis are cold sensitive [57]. Three mutants lacking one of the redundant ribosomal subunits displayed sensitivity to high pressure and low temperature (Figure 4). This is consistent with well-known knowledge that *E. coli* and *S. cerevisiae* mutants with defective ribosome assembly are cold sensitive [67,68,69,70]. The lack of Ccr4 and Pop2 (which comprise the Ccr4-Not transcriptional regulator [71]), is known to cause marked cold sensitivity; however, the underlying mechanism remains unknown [72]. The deletion of these genes also causes high-pressure sensitivity (Figure 4). Despite the determination of the transcriptional regulator, it is difficult to identify which target genes, of the Ccr4-Not complex, are responsible for high-pressure growth as numerous genes are under the control of this regulator. Although a global screening was performed to identify genes required for tolerance to freeze–thaw stress [73], no overlapping genes required for growth under cold or high-pressure environments [64], were identified. Therefore, a distinctive set of genes support these physiological characteristics of yeast. Deletion of genes involved in the later steps of ergosterol biosynthesis promoted the accumulation of sterol derivatives in cells [74]. Such mutations cause sensitivity to high pressure and low temperature [64]. Thus, ergosterol maintains membrane property by opposing the hardening of the membrane caused by high pressure and low temperature.

## 8. High Pressure Activates a Nutrient Sensor Protein Kinase Complex TORC1

The aforementioned global screening methods revealed that the deletion of genes encoding the EGO complex (*EGO1, EGO3, GTR1,* and *GTR2*) [64,75] results in severe growth deficiency under high pressure and low temperature (Figure 3 and Figure 4). The EGO complex is required to tether the target of rapamycin complex 1 (TORC1). TOR is an evolutionarily serine/threonine kinase in eukaryotes. It regulates the cellular homeostasis by coordinating anabolic and catabolic processes when nutrients are present [76,77,78,79,80]. TORC1 promotes cell growth by activating synthesis of proteins and ribosomes in the presence of favorable nutrients [81]. However, when the cells are starved or exposed to lethal stresses, TORC1 signaling is repressed, which leads to the downregulation of protein synthesis and effectively arrests cell growth [81,82,83,84]. A pressure of 25 MPa stimulated TORC1 to promote the phosphorylation of a downstream effector protein such as Sch9 [85]. This stimulation depends on the intracellular glutamine sensor protein, Pib2. Mutants with deleted EGOC complex have an aberrant increase in glutamine levels. Therefore, the EGOC–TORC1 complex plays a critical role in maintaining an appropriate glutamine level under high pressure by downregulating the *de novo* synthesis of glutamine [85]. Although the mechanism by which high pressure causes glutamate accumulation is unclear, these findings provide a unique framework for understanding metabolic adaptation to high pressure. Few reports in other organisms have addressed the amino acid requirement for growth under high pressure. One such report by Catio et al. demonstrated that piezo–hyperthermophilic archaeon *Thermococcus barophilus* requires only three amino acids (glutamate, cysteine, and tyrosine) for growth under atmospheric pressure at 85 °C in peptone replacement, whereas this organism required 17 amino acids (other than alanine, glutamine, and proline) for growth under 40 MPa [86]. Since the strain exhibits uptake activity of amino acids at 40 MPa, the increased requirement for amino acids cannot be explained by attenuation of substrate transport systems by high pressure. Instead, the authors suggested that there is a metabolic shift between atmospheric pressure and high pressure [86].

## 9. Identification of Novel *S. cerevisiae* Genes Required for Growth under High Pressure

Among the 84 gene deletion mutants that display growth deficiency under high hydrostatic pressure and/or low temperature, 24 deletion mutants can be reversed by the introduction of four plasmids (*LEU2, HIS3, LYS2,* and *URA3*) together that allow the growth at 25 MPa, thereby suggesting a close association between the genes and nutrient uptake [87]. Most highly ranked genes associated with high-pressure growth, including *MAY24*, are poorly characterized. May24 is localized in the endoplasmic reticulum (ER) membrane. Therefore, the gene is designated as *EHG1* (ER-associated high-pressure growth gene 1) [87]. The deletion of *EHG1* caused decreases in nutrient transport rates and reduced nutrient permease levels when the cells were cultured at 25 MPa. Thus, Ehg1 is required for the stability and functionality of permeases under high pressure (Figure 3 and Figure 4) [65]. Ehg1 physically interacted with the histidine permease Hip1, leucine permease Bap2, and uracil permease Fur4. By functioning as a novel chaperone that facilitates coping with high-pressure-induced perturbations, Ehg1 could exhibit a stabilizing effect on nutrient permeases when they are present in the ER [87]. The acquisition of nutrients is one of the crucial biological processes in organisms under normal or harsh environmental conditions. The poorly characterized proteins including Ehg1 play a role in ensuring nutrient uptake via stabilizing the permeases present in the plasma membrane.

## 10. Conclusions

The use of genetic databases and functional genomic screening of *S. cerevisiae* can improve our fundamental understanding of the effects of high hydrostatic pressure on living cells. Many yeast strains belonging to the genera *Candida* and *Debaryomyces* are found in deep-sea environments and share many common genes with *S. cerevisiae*, which is not found in deep-sea environments. This review summarizes the mechanisms and biological processes associated with high-pressure adaptation in eukaryotes, especially yeasts.

Hydrostatic pressure is a thermodynamic variable that solely affects the equilibrium or the rate associated with any chemical reactions, not introducing any composing elements into the experimental system. The system reverts to its original state after decompression; hence, hydrostatic pressure can be utilized as a cleaning tool to modulate intracellular biochemical processes and may be applied for the production of any useful substances while saving energy. In the studies on “piezophysiology,” high hydrostatic pressure is used as a variable to elucidate biological processes accompanying considerable structural changes in living cells. However, we continue to know very little about the effects of high pressure on complex intracellular networks or how to deal with cellular responses to high pressure. A mechanistic understanding of the effects of high pressure on proteins, membranes, and small intracellular molecules is necessary to understand the complete mechanism of cellular responses to high pressure. Advanced transcriptomics, proteomics, and metabolomics studies of yeast under high pressure will be helpful in the selection of yeast strains, media, temperature, and holding time.

## Figures and Tables

**Figure 1 biology-10-01305-f001:**
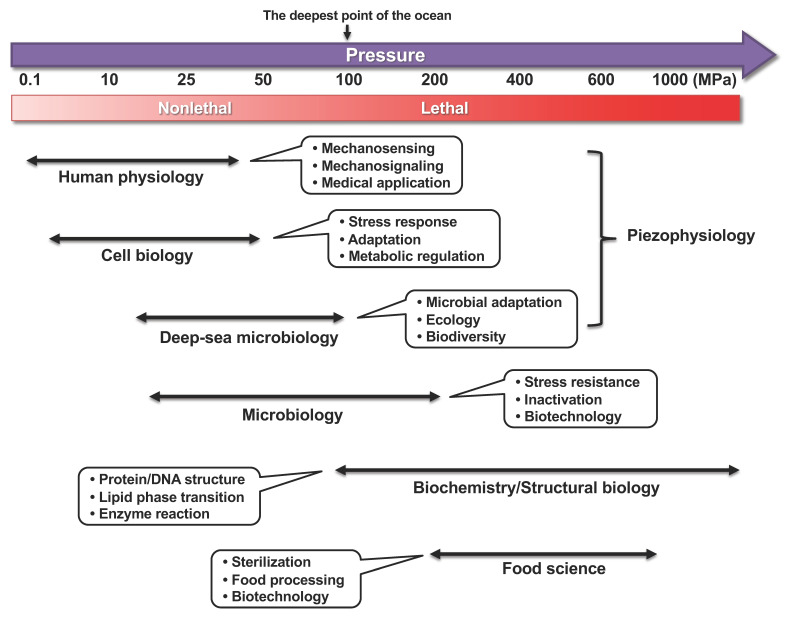
Pressure ranges, research fields, and main subjects in high-pressure bioscience and biotechnology. Thin double-headed arrows indicate approximate pressure ranges for corresponding research. The lethality of organisms under high pressure greatly depends on the species and duration of applied pressure and temperature.

**Figure 2 biology-10-01305-f002:**
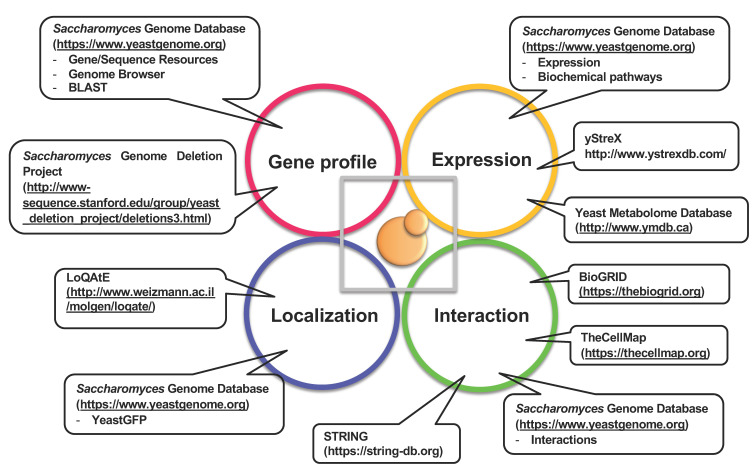
Bioinformatic tools for studying *S. cerevisiae* as a model to elucidate the mechanisms of high-pressure responses and adaptation.

**Figure 3 biology-10-01305-f003:**
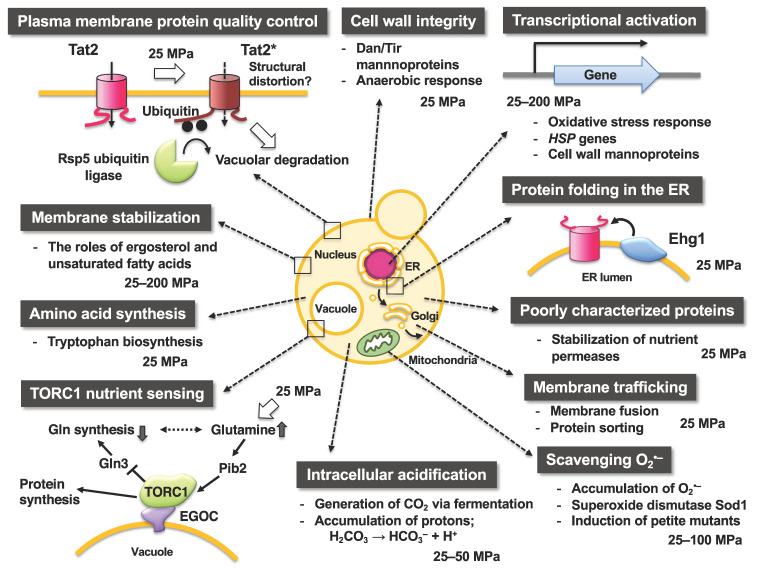
Occurrences and significance of intracellular changes in yeast responding to high hydrostatic pressure. Note that the figure only depicts limited aspects of the effects of high pressure on cellular functions in *S. cerevisiae*.

**Figure 4 biology-10-01305-f004:**
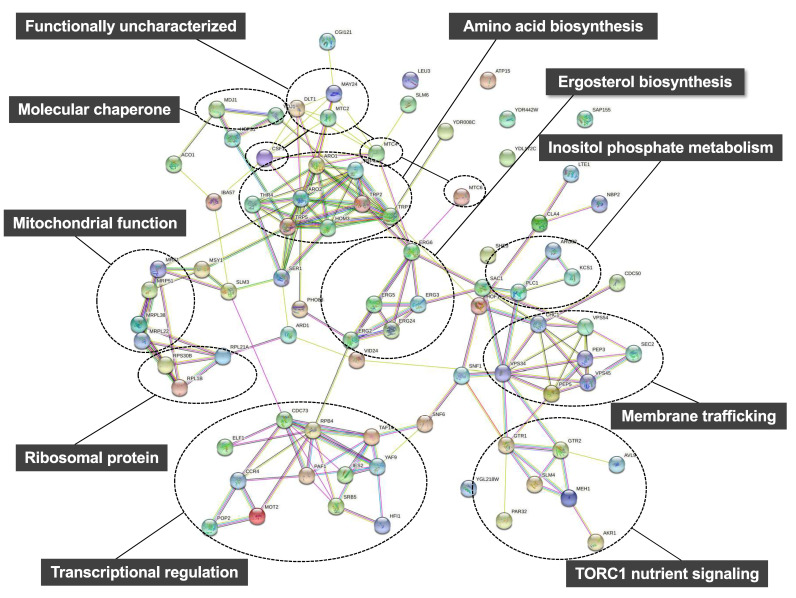
Prediction of protein-protein interactions among the 84 proteins encoded by high-pressure (25 MPa, 24 °C) and/or low-temperature (0.1 MPa, 15 °C) growth genes on STRING including direct (physical) and indirect (functional) associations (https://string-db.org; accessed on 8 December 2021).

**Table 1 biology-10-01305-t001:** Approximate pressure ranges affecting various biological processes.

Cellular Function	Inhibitory/Effective Pressure (MPa<)
Nutrient uptake	10
Cell division	20
Alcohol fermentation	15–20
Membrane protein function	25–50
DNA replication	25–50
RNA transcription	50–100
Protein synthesis	50
Microbial death	100–200
Protein oligomerization	50–100
Soluble enzyme activity	100
Protein tertiary structure	200–1000
DNA double strand formation	1000

**Table 2 biology-10-01305-t002:** Biological processes (gene ontology) for proteins required for growth under high pressure and/or low temperature.

Term Description	Observed Gene Count	Background Gene Count	Strength
Lysosome organization	3	3	1.9
Tryptophan biosynthetic process	4	6	1.73
Inositol phosphate biosynthetic process	3	5	1.68
Aromatic amino acid family biosynthetic process	6	24	1.3
Ergosterol biosynthetic process	5	25	1.2
Alcohol biosynthetic process	8	54	1.07
Positive regulation of transcription elongation from RNA polymerase II promoter	6	46	1.02
Organic hydroxy compound biosynthetic process	9	76	0.97
Transcription elongation from RNA polymerase II promoter	6	55	0.94
Cellular amino acid biosynthetic process	10	131	0.78
Small molecule biosynthetic process	19	324	0.67
Carboxylic acid biosynthetic process	11	186	0.67
Cellular amino acid metabolic process	12	246	0.59
Positive regulation of cellular biosynthetic process	16	424	0.48
Organic cyclic compound biosynthetic process	30	931	0.41
Small molecule metabolic process	21	693	0.38
Aromatic compound biosynthetic process	25	871	0.36
Organic substance biosynthetic process	46	1810	0.31
Cellular biosynthetic process	44	1764	0.3
Cellular nitrogen compound biosynthetic process	31	1261	0.29

The 84 genes required for *Saccharomyces cerevisiae* cell growth under high pressure (15 MPa, 24 °C) and/or low temperature (0.1 MPa, 15 °C) were classified into biological processes using STRING, a tool for functional enrichment analyses (https://string-db.org; accessed on 8 December 2021). The observed gene count indicates how many proteins in the network are associated with a particular term. The background gene count indicates how many proteins in total (in the network and in the background) have this term assigned. The strength indicates Log_10_ (observed/expected), which describes how large the enrichment effect is.

## Data Availability

Not applicable.

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
