# Peer review of "Molecular Responses to High Hydrostatic Pressure in Eukaryotes: Genetic Insights from Studies on Saccharomyces cerevisiae"

_biology, 2021, doi:10.3390/biology10121305_

Round 1

Reviewer 1 Report

The review by Fumiyoshi Abe discusses the effect on high hydrostatic pressure on various aspects of cellular physiology. The content, scope, and the overarching message of the review fits well with the journal.

Strengths

  1. The review covers topics ranging from the structural changes in macromolecules including, proteins and lipids to gene expression responses to high pressure stress and the challenges associated in deciphering the biological responses to high pressure. The discussion of the cited literature is well carried out.
  2. The review cites previous work in the field from last 25 years, so provides the reader an historical perspective and breadth of research conducted in the field.

Weakness

Some comments on the organization of the review-

Major comments

  1. Line 169-180, the author can provide a general perspective on how these different treatments pre-condition the cells for piezotolerance.
  2. Line 184, For Msn2/4 transcription factor, the author can comment on the possible role of the cell wall genes, that are regulated by these transcription factors, in context of piezotolerance.
  3. In the light of the role of nutrient uptake, the nutrition-sensing pathway, TORC1, and metabolic regulation, the author can provide some discussion on what literature exists (or is lacking) on the nutritional requirements and the regulation of nutritional status/metabolic flexibility in piezophiles and certain mammalian tissues that consistently experience high pressure stress.

Minor comments

  1. Reference #11 is not accessible online
  2. The title of table 1 does not accurately represent the content of the table. Maybe change the title to “pressure ranges for various biological processes” or something similar
  3. Line 60, the citation for table 1 is mentioned as ref #12. However, the citation in the table title is mentioned as ref #11. Please rectify.
  4. Figure 3 on page 8 has been repeated on page 6 without figure number and figure legend.

Author Response

I would like to thank you for your valuable comments that allowed me to improve the manuscript.

Major comments

  1. Line 169-180, the author can provide a general perspective on how these different treatments pre-condition the cells for piezotolerance.

>Answer:

Thank you for your comment. The general perspective has been described in the section “Effects of lethal levels of high pressure on the yeast survival” in the revised manuscript in terms of induction of oxidative stress responses referring additional publications.

  1. Line 184, For Msn2/4 transcription factor, the author can comment on the possible role of the cell wall genes, that are regulated by these transcription factors, in context of piezotolerance.

>Answer:

Thank you for your comment. I have stated that the DAN/TIR cell wall mannoprotein genes are not under the control of Msn2/4 transcription factor in the section “Effects of lethal levels of high pressure on the yeast survival” in the revised manuscript.

  1. In the light of the role of nutrient uptake, the nutrition-sensing pathway, TORC1, and metabolic regulation, the author can provide some discussion on what literature exists (or is lacking) on the nutritional requirements and the regulation of nutritional status/metabolic flexibility in piezophiles and certain mammalian tissues that consistently experience high pressure stress.

>Answer:

Thank you for your comment. To the best of my knowledge, there is no report other than our study (Ref. 84) addressing the TORC1 functionality on intracellular amino acid homeostasis under high pressure. Few reports in other organisms have addressed the amino acid requirement for growth under high pressure. One such report by Catio et al. demonstrated that piezo-hyperthermophilic archaeon Thermococcus barophilus requires only three amino acids (glutamate, cysteine, and tyrosine) for growth under atmospheric pressure at 85 °C in peptone replacement, whereas this organism required 17 amino acids, other than alanine, glutamine, and proline, for growth under 40 MPa. This has been described in the section “High pressure activates a nutrient sensor protein kinase complex TORC1”.

Minor comments

  1. Reference #11 is not accessible online

>Answer:

Thank you. I assume that it will be available online soon.

  1. The title of table 1 does not accurately represent the content of the table. Maybe change the title to “pressure ranges for various biological processes” or something similar.

>Answer:

Thank you for your comment. It has been changed to “Approximate pressure ranges affecting various biological processes”.

  1. Line 60, the citation for table 1 is mentioned as ref #12. However, the citation in the table title is mentioned as ref #11. Please rectify.

>Answer:

Thank you, it was my mistake.

  1. Figure 3 on page 8 has been repeated on page 6 without figure number and figure legend.

>Answer:

Thank you, it was an error in the editorial process.

Reviewer 2 Report

This paper describes the effect of high pressure on various living organisms from yeast to human. This is an excellent review form the expert of the high pressure microbiology.

The paper needs only small corrections.

  1. Page 1 row 40 "kg of force" is a strange expression, since kg is the unit of the mass, so weight of 1 kg mass could be the scientifically correct expression. I would recommend to leave about the "=1.0197 kg of force/cm2"
  2. 1. The author should distinguish between the uniaxial stress, and the isostatic pressure. Uniaxial stress can act on the bones, while isostatic pressure (which follows the Pascal-law, being isotropic, acts equally in all directions) can be experienced in the deep see, or during diving. I recommend to add few sentences in the first section to clarify this.
  3. Figure 1 shows typical pressures. However, the effect of the pressure depends very much on the actual object. E.g. protein structures can remain intact above much higher pressures as indicated here. GFP, or heat shock proteins can be stable even above 1000MPa (Scheyhing, C.H., et al., Biopolymers, 2002. 65(4): p. 244-253., Tolgyesi, F. et al.  Cellular and Molecular Biology 508 50, 361-369 (2004).) Also the sentence on page 5 row 117 gives a quite low pressure range. Even myoglobin is stable up to 500-600 MPa.
  4. Page 5 . The pressure applied fo meniscus tissue in the described experiment should be given (rows 138-139).
  5. The Figure caption on Page 6 is missing.
  6. Actually the same figure can be found on Page 8 (with caption) and Page 6 (without caption).
  7. The reviewer remarks that all the figures are taken from an earlier paper of the author (ref 11). Hopefully there is not much overlap between the present article and ref 11.
  8. The paper starts like a general review, but later, more than the half of it seems to deal with the yeast. This makes the paper a bit unbalanced. May be the author should not hide his intention to write mainly about the yeast which is his specialty. A less general title could express this. The first sections describing the general effect of pressure (including pressure on humans) can remain in the manuscript of course as an introduction.

I think the paper will be acceptable after these minor corrections.

Author Response

I would like to thank you for your valuable comments that allowed me to improve the manuscript.

The paper needs only small corrections.

  1. Page 1 row 40 "kg of force" is a strange expression, since kg is the unit of the mass, so weight of 1 kg mass could be the scientifically correct expression. I would recommend to leave about the "=1.0197 kg of force/cm2"

>Answer:

Thank you for your comment. "=1.0197 kg of force/cm2" has been deleted in the revised manuscript.

  1. 1. The author should distinguish between the uniaxial stress, and the isostatic pressure. Uniaxial stress can act on the bones, while isostatic pressure (which follows the Pascal-law, being isotropic, acts equally in all directions) can be experienced in the deep see, or during diving. I recommend to add few sentences in the first section to clarify this.

>Answer:

Thank you for your comment. I have stated as follows in the section “General effects of high hydrostatic pressure on biological systems”: While high hydrostatic pressure is a commonly known characteristic of deep-sea environments, the human body also experiences high pressure. However, it is important to distinguish between the isostatic pressure acting equally in all directions and the uniaxial stress. While deep-sea organisms are constantly exposed to the isostatic pressure, uniaxial, or directional, pressure can act on human bones. This review primarily focuses biological responses to isostatic hydrostatic pressure.

  1. Figure 1 shows typical pressures. However, the effect of the pressure depends very much on the actual object. E.g. protein structures can remain intact above much higher pressures as indicated here. GFP, or heat shock proteins can be stable even above 1000MPa (Scheyhing, C.H., et al., Biopolymers, 2002. 65(4): p. 244-253., Tolgyesi, F. et al.Cellular and Molecular Biology 508 50, 361-369 (2004).) Also the sentence on page 5 row 117 gives a quite low pressure range. Even myoglobin is stable up to 500-600 MPa.

>Answer:

Thank you for your comment. I have revised the statement referring the two papers you kindly suggested and extended the double-headed arrow to 1000 MPa in Fig. 1

  1. Page 5 . The pressure applied fo meniscus tissue in the described experiment should be given (rows 138-139).

>Answer:

Thank you for your comment. I have thoroughly revised the section.

  1. The Figure caption on Page 6 is missing.

>Answer:Thank you, it was an error in the editorial process.

  1. Actually the same figure can be found on Page 8 (with caption) and Page 6 (without caption).

>Answer:

Thank you, I have corrected it in the revised manuscript.

  1. The reviewer remarks that all the figures are taken from an earlier paper of the author (ref 11). Hopefully there is not much overlap between the present article and ref 11.

>Answer:

Thank you for your comment. I have dropped all figures from the original manuscript. Instead, I created new Figs. 1–4 for the revision.

  1. The paper starts like a general review, but later, more than the half of it seems to deal with the yeast. This makes the paper a bit unbalanced. May be the author should not hide his intention to write mainly about the yeast which is his specialty. A less general title could express this. The first sections describing the general effect of pressure (including pressure on humans) can remain in the manuscript of course as an introduction.

>Answer:

Thank you for your comment. I have referred additional 27 articles published from others (not from our group) in the revised manuscript indicative of general perspectives. However, studies on S. cerevisiae are still the mainstream of the present review. Therefore, I changes the title to: Molecular responses to high hydrostatic pressure in eukaryotes: genetic insights from studies on Saccharomyces cerevisiae

Reviewer 3 Report

The manuscript from Abe reviewed the effect of high hydrostatic pressure on eukaryotic cell responses, focusing on yeast S. cerevisiae species. It is interesting to the readers the genetic responses of yeast to the high hydrostatic pressure reviewed in the paper. Some comments I have are:

  1. There is one paragraph of “Simple summary” and one paragraph of “Abstract” on the first page. Unless otherwise required as a format, these two sections are too repetitive, and one abstract paragraph alone would serve the purpose.
  2. The five figures in this articles are all reprints from a recently published review written by the same author: “The effects of high hydrostatic pressure on the complex intermolecular networks in a living cell. The Review of High Pressure Science and Technology 31, 54-65.” The original review is in Japanese. It is a concern that the current article under review contains the exact same figures with an already published review. Even though the author cited the published review here, it is too much replication of published content unless the author can justify this reproduction.
  3. Figure 3 appeared twice on page 6 and on page 8. I believe the one on page 6 is a mistake insertion.
  4. The author is encouraged to include more citations in the reference, especially from other people's work in general.

Author Response

I would like to thank you for your valuable comments that allowed me to improve the manuscript.

  1. There is one paragraph of “Simple summary” and one paragraph of “Abstract” on the first page. Unless otherwise required as a format, these two sections are too repetitive, and one abstract paragraph alone would serve the purpose.

>Answer:

Thank you for your comment. I agree with you but I have learned that “Simple summary” is mandatory for publication in this journal.

  1. The five figures in this articles are all reprints from a recently published review written by the same author: “The effects of high hydrostatic pressure on the complex intermolecular networks in a living cell. The Review of High Pressure Science and Technology 31, 54-65.” The original review is in Japanese. It is a concern that the current article under review contains the exact same figures with an already published review. Even though the author cited the published review here, it is too much replication of published content unless the author can justify this reproduction.

>Answer:

Thank you for your comment. I have dropped all figures from the original manuscript. Instead, I created new Figs. 1–4 for the revision.

  1. Figure 3 appeared twice on page 6 and on page 8. I believe the one on page 6 is a mistake insertion.

>Answer:

Thank you, it was an error in the editorial process.

  1. The author is encouraged to include more citations in the reference, especially from other people's work in general.

>Answer:

Thank you for your comment. I have referred additional 27 publications from other groups to include more of general achievements such as the role of TMAO in deep-sea fishes.

Round 2

Reviewer 3 Report

The author modified the figures as suggested. The revised manuscript is improved and warrants publication in Biology.